# The Virus-Induced Upregulation of the miR-183/96/182 Cluster and the FoxO Family Protein Members Are Not Required for Efficient Replication of HSV-1

**DOI:** 10.3390/v14081661

**Published:** 2022-07-28

**Authors:** Andreja Zubković, Ines Žarak, Ivana Ratkaj, Filip Rokić, Maja Jekić, Marina Pribanić Matešić, Ricardo Lebrón, Cristina Gómez-Martín, Berislav Lisnić, Vanda Juranić Lisnić, Stipan Jonjić, Dongli Pan, Oliver Vugrek, Michael Hackenberg, Igor Jurak

**Affiliations:** 1Department of Biotechnology, University of Rijeka, 51000 Rijeka, Croatia; andreja.zubkovic@biotech.uniri.hr (A.Z.); ines.zarak@ugent.be (I.Ž.); ivana.ratkaj@biotech.uniri.hr (I.R.); maja.badurina@medri.uniri.hr (M.J.); marina.pribanic.matesic@medri.uniri.hr (M.P.M.); 2Laboratory for Advanced Genomics, Rudjer Boskovic Institute, 10000 Zagreb, Croatia; frokic@gmail.com (F.R.); ovugrek@irb.hr (O.V.); 3Genetics Department & Biotechnology Institute, Biomedical Research Center (CIBM), University of Granada, 18100 Granada, Spain; rlebron88@gmail.com (R.L.); cris12gm@gmail.com (C.G.-M.); hackenberg@go.ugr.es (M.H.); 4Center for Proteomics, Faculty of Medicine, University of Rijeka, 51000 Rijeka, Croatia; berislav.lisnic@medri.uniri.hr (B.L.); vanda.juranic@medri.uniri.hr (V.J.L.); stipan.jonjic@medri.uniri.hr (S.J.); 5Department of Medical Microbiology and Parasitology, Zhejiang University School of Medicine, Hangzhou 310058, China; pandongli@zju.edu.cn

**Keywords:** herpes simplex virus 1, HSV-1, virus-host interaction, miRNA, FoxO

## Abstract

Herpes simplex virus 1 (HSV-1) expresses a large number of miRNAs, and their function is still not completely understood. In addition, HSV-1 has been found to deregulate host miRNAs, which adds to the complexity of the regulation of efficient virus replication. In this study, we comprehensively addressed the deregulation of host miRNAs by massive-parallel sequencing. We found that only miRNAs expressed from a single cluster, miR-183/96/182, are reproducibly deregulated during productive infection. These miRNAs are predicted to regulate a great number of potential targets involved in different cellular processes and have only 33 shared targets. Among these, members of the FoxO family of proteins were identified as potential targets for all three miRNAs. However, our study shows that the upregulated miRNAs do not affect the expression of FoxO proteins, moreover, these proteins were upregulated in HSV-1 infection. Furthermore, we show that the individual FoxO proteins are not required for efficient HSV-1 replication. Taken together, our results indicate a complex and redundant response of infected cells to the virus infection that is efficiently inhibited by the virus.

## 1. Introduction

Viruses induce the extensive deregulation of cellular biological processes and exploit their metabolism for efficient replication and spread. Such deregulation can include various strategies to optimize energy consumption (reviewed in [1]), the subversion of defense mechanisms (reviewed in [2]), or signaling pathways (reviewed in [3]). It has also been observed that virus infection inevitably leads to perturbation of the host microRNAs (miRNAs), which might affect any of the mentioned processes. miRNAs are small non-coding RNA molecules that post-transcriptionally regulate the expression of genes by pairing with the complementary nucleotides, usually within the three prime untranslated regions (3′UTR) of mRNAs [4]. The nucleotides from 2 to 8 of the miRNAs have an essential role in binding specificity to the target sites [5]. Multiple studies have unveiled that viruses utilize host miRNAs to control a host as well as their own gene expression. For instance, the hepatitis C virus (HCV) shows a unique connection with miR-122, abundantly expressed in the liver, which primes the environment for efficient virus proliferation and persistence within the host [6]. In particular, large DNA viruses such as herpes simplex virus 1 (HSV-1) have been found to encode their miRNAs, in addition to exploiting those of the host [7,8,9]. HSV-1 is a widespread human pathogen known as the causative agent of cold sores. In rare cases, HSV-1 can also cause life-threatening infections and diseases such as encephalitis [10]. The primary site of infection is usually mucosal epithelium, where the virus enters the cells and abundantly expresses its genes in a controlled cascade of gene expression, first immediate-early (IE), early (E), and late proteins (L), finally leading to the generation of new virions. Additionally, HSV-1 can infect nearby innervating sensory neurons and travel retrogradely from the peripheral axons to the neuronal body to establish lifelong latent infection. In contrast to the productive infection, during latency, the virus does not replicate and the only abundantly expressed transcripts are those arising from the latency-associated transcript (LAT) locus, which gives rise to stable long non-coding RNAs (lncRNA) called LAT intron, and a number of microRNAs [11]. 

Occasionally, HSV-1 can reactivate and travel anterogradely to the periphery to re-enter the productive phase of infection and spread [10]. HSV-1 has been shown to express 29 mature miRNAs (vmiRNAs), all of which can be detected, at various levels, during the productive phase of infection. However, only a small subset of vmiRNAs has been detected in latently infected human neurons [12,13] or latency models [14]. This includes vmiRNAs arising from the LAT region (miR-H2, -H3, -H4, -H5, -H7, and -H8) and miRNAs located just upstream of the LAT transcription start site (miR-H1 and miR-H6). The exact roles of vmiRNAs in HSV-1 infection are yet to be revealed, nonetheless, for some, a role in the control of latency had been demonstrated [11,15,16,17,18,19]. On the other hand, several studies have reported the deregulation of host miRNAs after HSV-1 infection [20], and for some, their functional importance has been proposed. For example, miR-23a, miRNA upregulated after HSV-1 infection, decreases the levels of interferon regulatory transcription factor 1 (IRF1), which in turn results in reduced levels of viperin, an important anti-viral gene, and thus facilitates virus infection [21,22]. Likewise, it has been shown that miR-132 is induced in the KSHV infection of lymphatic endothelial cells (LECs) as well as HSV-1 or HCMV infection of monocytes and it targets p300, a protein that associates with CREB and is an important mediator of antiviral immunity. By decreasing the levels of p300, the expression of IFN-β, ISG15, IL-1β, and IL6 is impaired, resulting in the suppression of antiviral immunity, thus facilitating viral replication [23]. Interestingly, miR-138, which is abundantly present in neurons, limits the expression of ICP0, an important IE viral protein crucial for efficient reactivation, thereby repressing the productive cycle and promoting latency in mice [24]. Additionally, miR-138 has been found to decrease the expression of host genes important for virus replication, transcription factors Oct-1 and FoxC1, to create a suitable environment for latent infection [25].

The miRNA cluster miR-183/96/182, which is transcribed as a single pri-miRNA transcript and further processed into three separate pre-miRNAs, has been shown to be deregulated in cells infected with different herpesviruses including HSV-1, human cytomegalovirus (HCMV), and Epstein–Barr virus (EBV) [26,27,28]. In HSV-1 infected cells, this cluster, which is suppressed by host protein ZEB, is regulated by ICP0, which directs the host protein ZEB for ubiquitin-dependent proteasomal degradation, which in turn leads to the de-repression of the miR-183/96/182 cluster [29]. However, the biological relevance of this deregulation is not known.

In this study, we comprehensively investigated the host miRNA expression during productive HSV-1 infection using massive parallel sequencing of miRNAs coupled with transcriptome analysis. We report that only miRNAs transcribed from the single cluster, miR-183/96/182, were reproducibly and significantly upregulated in HSV-1 infected human foreskin fibroblasts (HFFs). These miRNAs target the Forkhead box O (FoxO) family of transcription factors tightly regulated by the insulin/PI3K/Akt signaling pathway. We show that individual FoxO family protein members are not required for efficient HSV-1 infection and that miR-183, -96, and -182 have a minor role in the regulation of these proteins.

## 2. Materials and Methods

### 2.1. Cells and Viruses

Human embryonic kidney cells (HEK293, CRL-1573), epithelial cells of the African green monkey kidney (Vero, CCL-81), primary human foreskin fibroblast (HFF), primary mouse embryonic fibroblast (MEF, a generous gift of S. Jonjić, Faculty of Medicine, University of Rijeka, Rijeka, Croatia), and human lung fibroblast cells (WI-38, CCL-75) were grown in Dulbecco’s modified Eagle medium (DMEM; PAN-biotech GmbH, Aidenbach, Germany) supplemented with 10% fetal bovine serum (FBS; PAN-biotech GmbH, Aidenbach, Germany), penicillin/streptomycin 100 μg/μL, 2 mM L-glutamine (Capricorn Scientific, Ebsdorfergrund, Germany), and 1 mM sodium pyruvate (Capricorn Scientific, Ebsdorfergrund, Germany) at 37 °C in the presence of 5% CO_2_. Mouse bone marrow-derived macrophages (BMDM; a generous gift from I. Munitić, Department of Biotechnology, University of Rijeka, Rijeka, Croatia) were maintained in RPMI (Lonza, Basel, Switzerland) medium containing 10% of FBS, penicillin/streptomycin 100 μg/μL, and 2 mM L-glutamine. Human neuroblastoma cells (SH-SY5Y, CRL-2266) were maintained in MEM (Minimum Essential Medium) with Earle’s Salts (Capricorn Scientific, Ebsdorfergrund, Germany), supplemented with 15% FBS, penicillin/streptomycin 100 μg/μL, and 2 mM L-glutamine at 37 °C in 5% CO_2_. Human retinal pigment epithelial cells immortalized with hTERT (hTERT RPE-1, CRL-4000) were maintained in MEM with Earle’s Salts, supplemented with 10% FBS, penicillin/streptomycin 100 μg/μL, 2 mM L-glutamine, 0.5 mM sodium pyruvate, 15 mM HEPES (Capricorn Scientific, Ebsdorfergrund, Germany), 10 µg/mL hygromycin (MilliporeSigma, Burlington, MA, USA) at 37 °C in 5% CO_2_. HSV-1 strain KOS (kindly provided by DM. Coen, Harvard Medical School, Boston, MA, USA) was prepared and titrated in Vero cells as previously described [29] and stored at −80 °C. Mouse cytomegalovirus (MCMV) strain C3X (MCMV BAC pSM3fr cloned from MCMV Smith strain, ATCC VR-1399; kindly provided by Stipan Jonjić, Faculty of Medicine, University of Rijeka, Croatia) was prepared in MEF cells and infections were performed using centrifugal enhancement as previously described [30].

### 2.2. Infections and Reagents

Cells were seeded a day before the experiment and infected with HSV-1 or MCMV at the indicated MOI (multiplicity of infection) or MOCK infected (uninfected). One hour after infection, the infectious medium was replaced with a fresh growth medium. Samples for protein or nucleic acids analyses were collected at indicated hours post-infection (h.p.i.). As indicated, specific inhibitors or reagents were added 30 min before infection and maintained during the time course of the infection. High molecular weight polyinosinic:polycytidylic acid (poly [I:C]), 100 µg/mL (Invivogen, San Diego, CA, USA), and recombinant IFN-β (rIFN-β), 1 U/µL (PBL Biomedical Laboratories, Piscataway, NJ, USA), 20 µM MG132 (MilliporeSigma, Burlington, MA, USA), and 100 µM acyclovir (ACV; MilliporeSigma, Burlington, MA, USA), and puromycin (Carl Roth GmbH & Co. KG, Karlsruhe, Germany) were used according to the manufacturer’s instructions.

### 2.3. Plaque Assay

Virus titers were determined as previously described [29]. Briefly, confluent monolayers of Vero cells in 6-well plates were infected with serial 10-fold dilutions of the supernatant sampled from the HSV-1-infected cells and incubated with the infectious media for 1 h at 37 °C. The infectious medium was then removed and the cells were overlaid with a methylcellulose solution (3% FBS, DMEM, 100 μg/μL penicillin/streptomycin, 1mM Na-pyruvate in 1.2% methylcellulose), and incubated in 5% CO_2_ at 37 °C for 3 days. Cells were fixed using 5% (*v*/*v*) methanol/10% (*v*/*v*) acetic acid and stained with 5% Giemsa (Carl Roth GmbH & Co. KG, Karlsruhe, Germany) in PBS. The viral titers were determined as the number of plaque-forming units (PFU)/mL.

### 2.4. RNA Extraction and Quantitative RT-PCR (RT-qPCR)

The total RNA was extracted from MOCK and HSV-1 infected cells using TRI Reagent solution (Ambion, Inc., Austin, TX, USA), according to the manufacturer’s instructions. The total RNA samples were transcribed with the High Capacity cDNA Reverse Transcription Kit (Applied Biosystems, Waltham, MA, USA) and levels of the transcripts were determined by RT-qPCR using FastStart Essential DNA Green Master (Roche, Basel, Switzerland) according to the manufacturer’s instructions. The results were normalized to the expression of the 18S ribosomal RNA or mouse glyceraldehyde 3-phosphate dehydrogenase (GAPDH) and presented as the relative mRNA expression compared to 18S rRNA. The primers used were: FoxO1—forward, 5′-GCTTCCCACACAGTGTCAAGAC-3′; reverse, 5′-CCTGCTGTCAGACAATCTGAAGTAC-3′; FoxO3—forward, 5′-GGGGAACTTCACTGGTGCTA-3′; reverse; 5′-TGTCCACTTGCTGAGAGCAG-3′; 18S rRNA—forward, 5′-GTAACCCGTTGAACCCCATT-3′; reverse, 5′-CCATCCAATCGGTAGTAGCG-3′; GAPDH-mouse—forward, 5′-GGTGCTGAGTATGTCGTGGA-3′; reverse, 5′-GTGGTTCACACCCATCACAA-3′.

For the analysis of miRNAs, the total RNA was transcribed using the TaqMan microRNA Reverse Transcription Kit (Applied Biosystems, Waltham, MA, USA) according to the manufacturer’s instructions and used as a template in the following RT-qPCR reactions using the TagMan Universal PCR Master Mix (Applied Biosystems, Waltham, MA, USA). Specific TaqMan miRNA assays were used for the detection of human miR-23a-3p, miR-101-3p, miR-132-3p, miR-138-5p, miR-183-5p, miR-96-5p, miR-182-5p, let-7a-5p, and HSV-1 vmiR-H6-3p (Applied Biosystems, Waltham, MA, USA). All reactions were performed in biological triplicates using LightCycler 96 (Roche, Basel, Switzerland).

### 2.5. RNA Sequencing and Data Analysis

For the miRNA analysis, HFF cells were infected with HSV-1 and the total RNA was extracted at 8 h.p.i. and 18 h.p.i. Small RNA libraries were generated using the TrueSeq Small RNA Sample Preparation Kit (Illumina, Inc., San Diego, CA, USA) according to the manufacturer’s protocol and sequenced by single read sequencing (50 nt) using an Illumina HiSeq 2500 sequencer (Institute Ruđere Bošković, Laboratory for Advanced Genomics, Zagreb, Croatia). The total number of aligned sequence reads (read count) for each host and viral miRNA was normalized using read counts for human let-7a, the expression of which does not change during HSV-1 infection, or to the size of the library (RPM value), as previously described [14]. The sequence reads were analyzed with sRNAtoolbox (Computational Epigenomics Lab, Evolutionary Genomics and Bioinformatics Group. Dept. of Genetics, Inst. of Biotechnology, University of Granada, Spain), a versatile compilation of software for miRNA analysis [31,32], as previously described [20]. Briefly, microRNAs were mapped using Bowtie and miRbase sequence database (release 22) annotations [33]. Of note, the newly reported HSV-1 miRNAs, miR-H28 and miR-H29, were not represented within the miRbase but were included in the analysis. We looked for reads that did not contain mismatches in the first 19 nucleotides but accepted all sequences that started at most 3 nt upstream and ended at most 5 nt downstream of the reference sequence, and the minimum read length of the input reads was 15. All sequence reads detected at least once in any condition were used for differential expression. 

The transcriptome sequencing was performed using the TruSeq Stranded Total RNA Sample Preparation Guide (Illumina, Inc., San Diego, CA, USA), as previously described [34]. Briefly, samples were rRNA depleted and fragmented, primed for first- and second-strand cDNA synthesis, adenylated 3′ ends, adapters were ligated, and DNA fragments were enriched. The libraries were validated using the Agilent Bioanalyzer and sequenced on a NextSeq 500 Illumina platform at the Rudjer Boskovic Institute, Zagreb, Croatia. The sequencing quality was assessed using FastQC (Version 0.11.9, Babraham Bioinformatics, Babraham Institute, Cambridge, UK), and reads were mapped to the reference HSV-1 genome, strain KOS (GenBank accession number JQ673480.1), and the Homo sapiens (human) genome assembly GRCh38 (hg38) from the Genome Reference Consortium using TopHat (2.1.1 release, Center for Bioinformatics and Computational Biology, University of Maryland, College Park, MD, USA). Mapped reads were then visualized using Integrative Genomics Viewer (IGV; 2.10. release, Broad Institute, Cambridge, MA, USA, and the Regents of the University of California, Oakland, CA, USA). Finally, Cufflinks (2.2.1 release, Trapnell’s lab, University of Washington, Seattle, WA, USA) was used for differential expression and normalized expression levels were reported in FPKM (fragments per kilobase of exon per million fragments mapped).

### 2.6. Protein Extraction and Western Blot

To extract the proteins, cells were lysed in RIPA buffer (150 mM NaCl, 1% NP-40, 0.5% Na deoxycholate, 0.1% SDS, 50 mM Tris (pH 8.0), and protease inhibitors (cOmplete, Roche, Basel, Switzerland)), mixed with 2× Laemmli buffer with β-mercaptoethanol—(Santa Cruz Biotech, Dallas, TX, USA), and denatured for 6 min at 95 °C. Proteins were separated in 10% SDS–PAGE gels and transferred onto a nitrocellulose membrane (Santa Cruz Biotech, Dallas, TX, USA). Membranes were blocked in 5% *w*/*v* nonfat dry milk in 1× Tris-buffered saline (TBS) for 30 min at room temperature and incubated with the specified primary antibodies at 4 °C with gentle rotation overnight. The primary antibodies and dilutions used in the experiments were as follows: α-actin (MilliporeSigma, Burlington, MA, USA)—1:10000, α-gC (Abcam, Cambridge, UK)—1:2000, α-ICP0 (Abcam, Cambridge, UK)—1:2000, α-ICP4 (Abcam, Cambridge, UK)—1:2000, α/β tubulin (Cell Signaling Technology, Inc., Danvers, MA, USA)—1:1000, α-FoxO1 (Cell Signaling Technology, Inc., Danvers, MA, USA)—1:1000, α-FoxO3 (Cell Signaling Technology, Inc., Danvers, MA, USA)—1:1000, α-FoxO4/AFX-1 (Santa Cruz Biotech, Dallas, TX, USA)—1:100, α-P-FoxO1/3 (Cell Signaling Technology, Inc., Danvers, MA, USA)—1:1000, α-P-FoxO3 (Cell Signaling Technology, Inc., Danvers, MA, USA)—1:1000. Blots were washed for 30 min with TBS-0.05% Tween 20 (TBS-T) and primary antibodies were detected using horseradish peroxidase-conjugated goat α -rabbit or α -mouse secondary antibodies both diluted 1:2000 (Cell Signaling Technology, Inc., Danvers, MA, USA) and incubated at room temperature for 1 h. Blots were again washed for 30 min and visualized using the Amersham ECL reagent or SuperSignal West Femto Maximum Sensitivity Substrate (Thermo Fisher Scientific, Waltham, MA, USA) and ChemiDoc MP (Bio-Rad Laboratories, Hercules, CA, USA). 

### 2.7. CRISPR-Cas9 Constructs and Validation of Knockout (−/−) Cells

Three guide RNAs for each gene of interest were designed using the GPP sgRNA Designer Tool (Broad Institute, Cambridge, MA, USA) [35,36] based on the human reference GRCh38. The list of the guide RNA (gRNA) sequences and constructs used in the study can be found in Appendix A. Briefly, synthetic dsDNA oligonucleotides were directly cloned using BsmBI (New England Biolabs, Ipswich, MA, USA) into pLentiCRISPRv2 (Addgene, Watertown, MA, USA), as previously described [37], and the cloned sequence was confirmed by sequencing (Eurofins Scientific, Luxembourg, Luxembourg). The HEK293 cells were transfected with the generated plasmid, together with the lentiviral packaging system based on psPAX2 and pMD2.G [37] using Lipofectamine 2000 (Thermo Fisher Scientific, Waltham, MA, USA) according to the manufacturer’s instructions. Lentiviral vectors carrying individual guide RNAs were collected from the supernatant, and vectors targeting the same gene were pooled, filtered, and added with polybrene at the final concentration of 8 µg/mL (MilliporeSigma, Burlington, MA, USA) to the SH-SY5Y cells. Twenty-four hours after transduction, cells were washed in PBS and the selection media were added (complete medium supplemented with puromycin at 1 µg/mL). After establishing the resistant culture, the depletion of the targeted protein was confirmed by Western blot analysis.

### 2.8. Statistical Analysis

To analyze the statistical significance, the Student’s t-test was used (GraphPad Prism 8, GraphPad Software, Inc., San Diego, CA, USA). A level of *p* ≤ 0.05 was considered statistically significant. The level of statistical significance is marked with asterisks: *** *p* ≤ 0.001; ** *p* ≤ 0.01; * *p* ≤ 0.05. Non-significant samples were not highlighted on the graphs.

## 3. Results

### 3.1. miR-183/96/182 Cluster Is Upregulated in HSV-1 Infected Primary HFFs

To comprehensively investigate the deregulation of host miRNAs during productive HSV-1 infection and their potential biological functions, we compared two independent miRNA sequencing datasets. First, we infected the primary HFFs at an MOI of 5, extracted RNAs from mock-infected cells, and cells infected with HSV-1 strain KOS at 8 and 18 h post-infection (h.p.i.), and analyzed miRNA expression by massive parallel sequencing. The sequences obtained were analyzed using sRNAtoolbox. In each experiment, we detected about 1500 human miRNA expressed at a wide abundance range (i.e., from 0.02—279,598 reads per million (RPM)) (Appendix A). On the other hand, HSV-1 encoded miRNAs represented only a minor fraction of the total miRNA in the sample (i.e., 0.24% and 0.68% at 8 and 18 h.p.i., respectively) (Appendix A), which is in accordance with similar previous studies. Our differential expression analysis showed significant deregulation (fold change ≥ 2) of 134 human miRNAs at 18 h.p.i. (Figure 1A), however, only a small fraction of miRNAs were deregulated with a fold change ≥3. As expected, at 8 h.p.i., we did not observe a significant deregulation of miRNAs compared to the mock; nonetheless, a slight trend in the deregulation of the same miRNAs that were significantly deregulated at 18 h.p.i. can be observed. The second experiment represented one of our previous sequencing efforts, which was performed similarly (i.e., using the identical virus strain (KOS) infecting the matching cell type (HFFs)) under similar conditions (10 and 24 h.p.i.) and the same sequencing platform. Of note, the overall sequencing quality was comparable (i.e., total number of reads, the HSV-1 miRNA pattern, and the relative quantity) (Appendix A). We hypothesized that by comparing two unrelated sequencing experiments, we would be able to identify the reproducibly deregulated miRNAs in HSV-1 infections. Surprisingly, the overlap between these two experiments was rather limited (i.e., only four miRNAs (miR-182, -183, -96, and -375) were reproducibly upregulated and two miRNAs were downregulated (miR-29 and -27a) under all experimental conditions) (Appendix A). The fold change in the reproducibly deregulated miRNAs is shown in Figure 1B. These results suggest a strong dependence on the cells used in the experiments and may explain the observed discrepancies between different studies. Nonetheless, miRNAs expressed from a single transcriptional cluster, miR-183, -96, and -182 [38] were reproducibly upregulated and showed the highest level of deregulation, indicating their importance. The same cluster was found upregulated in a limited study on two in vitro HSV-1 latency models (rat superior cervical ganglia derived neurons quiescently infected with HSV-1 GFP-US11 (Patton strain) and HFF quiescently infected with HSV-1 KOS strain) [29], but also in cells infected with other herpesviruses [27,28]. Importantly, we detected only minor fluctuations (<1.5×) in the expression levels for several miRNAs that had previously been shown to be significantly deregulated in HSV-1 infected cells [21,23,26,29,39,40,41,42,43,44,45,46,47]. As above-mentioned, the use of different viruses, different cells, and conditions could explain this discrepancy. 

To additionally challenge the reproducibility of our sequencing results and to address the upregulation of the miR-183/96/182 cluster in different cells, we infected two primary cells (i.e., human foreskin fibroblasts HFFs and mouse bone marrow-derived macrophages BMDM), human lung fibroblasts WI38 cells), and two widely used transformed cell lines (i.e., human embryonic kidney cells HEK293 and human neuroblastoma cells SH-SY5Y) and analyzed the miRNA expression by the stem-loop-RT-qPCR at 8 and 18 or 24 h.p.i. In the analysis, we included an additional four host miRNAs, miR-23a-3p, miR-101-3p, miR-132-3p, and miR-138-5p, previously reported as deregulated and/or important for HSV-1 infection [21,24,40,43]. Of note, the expression of HSV-1 late gene product, miR-H6 served as the indicator of late viral gene expression. Surprisingly, only miR-182, miR-96, and miR-183 were upregulated more than >2×, in both the fibroblast cells (HFFs and WI38; ~10×) (Figure 2A and Appendix A), but not in other cells (Figure 2B and Appendix A). Moreover, we observed only minor deregulation for all of other miRNAs tested, previously reported as deregulated. Thus, we selected miR-183/96/182 as the main target for further functional analysis. 

### 3.2. miRNAs miR-183, -96, and -182 Regulate a Number of Common Targets

To determine the potential targets of miR-183, -96, and -182, we implemented TargetScan 7.1 (Bioinformatics and Research Computing, Whitehead Institute for Biomedical Research, Cambridge, MA, USA), and miRDB (Wang’s lab, Department of Radiation Oncology, Washington University School of Medicine, St. Louis, MO, USA), two widely used algorithms that predict mRNA targets [48,49]. The prediction revealed a large number of highly conserved potential targets for each of these miRNAs (between 300 and 778 targets found by both prediction tools for each of the miRNAs; not shown). Notably, only 33 potential targets were identified as shared by all three miRNAs, and these largely represent genes encoding proteins involved in the regulation of gene expression (Figure 3A and Appendix A). Among these, *FoxO1* was identified as a potential target for all three miRNAs, and two other members of the Forkhead box protein family, *FoxO3* and *FoxO4*, were found as potential targets for miR-182 and miR-96, and solely miR-96, respectively (Figure 3A). The binding sites for each miRNA of the cluster predicted by TargetScan are shown in Appendix A. Members of the FoxO protein family are well-known transcriptional factors that are regulated by the insulin/PI3K/Akt signaling pathway and are known to have many diverse roles in cell cycle progression, apoptosis, metabolism, differentiation, and oxidative stress resistance [50,51,52,53] including roles in viral infections [54,55,56,57,58,59,60,61,62,63,64,65]. The regulation of the *FoxO* family by miRNAs of the mir-183/96/182 cluster is well-established and has been investigated in various cellular pathways including the modulation of the innate immune response in infections or autoimmune diseases, or in various cancers [66,67]. Thus, we did not additionally validate this regulation in this study. 

### 3.3. HSV-1 Induces the Expression and Post-Translational Modifications of FoxO Protein Family Members

Based on the target prediction, we hypothesized that the increase in miRNAs miR-96, -182, and -183 will consequently lead to decreased levels of the FoxO family members. To address this, we infected various cells with HSV-1 at an MOI of 1 and collected samples for the Western blot and RNA analysis at different time points after infection. We observed an initial increase in the FoxO1 and FoxO3 levels between 1 h.p.i. and 12 h.p.i. and a slight decrease later in infection (Figure 3B and Appendix A), which could be explained by the increase in miRNAs late in infection (Figure 2A). FoxO4 levels were steadily increased through the time of infection (Figure 3B). Of note, the same dynamic pattern of the FoxO proteins (first upregulation followed by downregulation) was observed in all cell lines tested (HEK293, RPE1, HFF, SH-SY5Y; Figure 3B and Appendix A). However, surprisingly, we observed FoxO1 and FoxO3 downregulation in cells in which we did not observe upregulated levels of miR-183/96/182 during the time course of infection (e.g., HEK293 (Appendix A)), indicating additional mechanisms other than miRNAs for the depletion of FoxO family members late in infection. Moreover, in cells infected and treated with different inhibitors including cycloheximide (CHX, protein synthesis inhibitor) and actinomycin D (ActD, RNA synthesis inhibitor), or in cells infected with UV inactivated virus, we did not observe the downregulation of FoxO proteins (Figure 3C). These results indicate that IE/E viral gene expression may be required for the depletion of FoxO proteins. Interestingly, infection in the presence of acyclovir (i.e., a condition that allows for the expression of IE but prevents the expression of L proteins), the depletion was not fully achieved (Figure 3B). On the other hand, the addition of a proteasome inhibitor (MG132) 30 min prior to infection (*), but not 2 h.p.i. (**), successfully prevented the depletion (Figure 3C). Taken together, these results may point to the role of the IE protein ICP0, a well-characterized ubiquitin ligase, but also to additional mechanisms such as miRNAs and late viral functions.

Furthermore, by carefully analyzing the expression of FoxO protein members by Western blot, we observed the appearance of double bands (i.e., a shift of a few kilodaltons (kDa) during the time course of infection (Figure 3B)), which might represent phosphorylated forms of proteins (i.e., inactive forms). Of note, this phenomenon was observed in most of the cell lines tested, but was particularly apparent in RPE cells (representative Figure 3B and Appendix A). Indeed, using phospho-specific antibodies, we confirmed the phosphorylation of FoxO1 and FoxO3. Interestingly, the addition of acyclovir, an inhibitor of viral DNA replication and late gene expression, prevented the shift of a few kDa of FoxO proteins (Figure 3B). The observed phosphorylation is consistent with the previous study by Chuluunbaatar et al., who found that FoxO1 can be phosphorylated by Akt kinase and viral kinase Us3.

Taken together, we show that the virus is equipped with multiple mechanisms to control the expression and function of FoxO proteins and that miRNAs might have only a minor contribution to this. 

### 3.4. Poly (I:C), IFN-β Treatment, and Infection with Other Herpesviruses Increase the Level of FoxO1 and FoxO3 Proteins

We were intrigued by the increased levels of proteins of the FoxO family in cells infected with HSV-1, which indicates the possible transcriptional activation of *FoxO* genes. To address this, we analyzed the transcriptome of HSV-1 infected cells. Briefly, we infected HEK293 cells with strain KOS at MOI 10 and collected the total RNA at 1, 4, 8, and 12 h.p.i. from two biological replicates for each time point. The sequencing libraries were prepared using the TruSeq Stranded Total RNA Sample Preparation Kit and sequenced on the NextSeq 500 Illumina platform. The transcriptome analysis showed an increase in *FoxO1* and *FoxO3* mRNAs (Figure 3D) early in infection (4 h.p.i.), however, the results were somewhat inconclusive for the *FoxO4* transcript due to the low number of the reads (not shown). The transcriptional activation of the *FoxO* family was additionally confirmed in a time-course experiment (Figure 3E) using RT-qPCR for *FoxO1* and *FoxO3*. Similar to the sequencing results, we observed an increase early after infection and a sharp drop in mRNA levels after 8 h.p.i. 

To further investigate whether the FoxO family induction is limited to HSV-1 infection or if it might represent a conserved host–response mechanism to infection with other herpesviruses, we studied the expression of FoxO members in cells infected with murine cytomegalovirus (MCMV). In brief, primary mouse embryonic fibroblasts (MEFs) were infected with the MCVM strain C3X, and samples for protein and RNA analysis were collected at different times post-infection. Interestingly, we observed a strong upregulation of all of the tested members of the FoxO protein family (Figure 4A) during the time course of MCMV infection. Moreover, the analysis of RNA confirmed the transcriptional activation of *FoxO* genes (Figure 4B), similar to HSV-1. It is important to note that although we observed an apparent increase in the phosphorylation of FoxO proteins in contrast to the HSV-1 infected human cells, we did not observe an obvious size shift (Figure 3B). At this point, we cannot explain this difference, but it might be related to the specificity of the antibodies or different activities of the virus and/or cellular kinases. 

Taken together, these results led us to the hypothesis that FoxO family members might be involved in intrinsic antiviral response and triggered in response to virus infection. For example, it has been found that FoxO1 promotes the degradation of IRF3 and limits *IRF7* transcription [69], while FoxO3 has been found to target *IRF7*, and thus, negatively regulate virus-induced type I interferon (IFN) expression [70,71,72]. To test this hypothesis, we analyzed the expression of FoxO1 and FoxO3 in response to the immunostimulants poly (I:C) and IFN-β, which mimic viral infection and elicit an antiviral response, respectively (Figure 4C). Indeed, the stimulation of RPE1 cells with poly (I:C) resulted in the gradual induction of both the FoxO protein levels, with a peak protein expression at 6 h post-stimulation. On the other hand, stimulation with IFN resulted in a sharp, but very transient, increase in FoxO1 and FoxO3 immediately after stimulation. The observed drop in the FoxO protein levels 3 h or 6 h after stimulation with IFN-β is in agreement with the work of Litvak et al., showing that IFN-I limits the transcription of FoxO3, which is responsible for targeting *IRF7*, and thus increases the transcription of *IRF7* for the maximum antiviral response [70]. Regardless, the results indicate that FoxO family proteins are induced as a result of an immune response to viral infection.

### 3.5. FoxO1 and FoxO3 Are Not Required for Efficient Replication of HSV-1

To address the biological relevance of the miR-183/96/182 cluster upregulation in HSV-1 infection on one hand, and the upregulation of FoxO family members on the other, we performed several functional assays. First, we asked whether the increased levels of miRNAs of the miR-183/96/182 cluster affected the virus replication. To test this, we transfected HFFs with mimics of miR-96, miR-182, and miR-183 or negative control mimics (NC), and 24 post-transfection, we infected cells with HSV-1 at a MOI of 5 (Figure 5A) and 0.001 (Figure 5B). To our surprise, we did not observe any obvious difference in virus replication (Figure 5A,B). Similarly, these miRNAs only had minor effects on viral infection in Neuro-2a cells using a high throughput assay in which cells were transfected with individual miRNAs and infected with HSV-1-luciferase-reporter virus (Pan et al., not published; personal communication). These results led us to the conclusion that the upregulation of the miR-183/96/182 cluster has, at best, a minor role in HSV-1 replication in cultured cells. 

Next, to investigate the role of FoxO proteins in HSV-1 infection, we inactivated the *FoxO1*, *Foxo3*, and *FoxO4* (*FoxO1 −/−*, *FoxO3 −/−*, *FoxO4 −/−*) genes in the SH-SY5Y cell using CRISPR-Cas9 technology. Briefly, we generated guides using the GPP sgRNA Designer Tool [35,36] and selected the top three different candidate guides to target each of the genes. Several individual clones were analyzed for the expression of targeted genes and only the complete knock-down cells were used in further experiments. We successfully knocked out FoxO1 and FoxO3 proteins (a representative Western blot Figure 5C); however, we were not able to generate FoxO4 deficient cells, or cells deficient for all FoxO proteins (not shown). Next, FoxO1 and FoxO3 deficient cells were infected at high and low MOIs, and virus replication was monitored through the time course of infection. Interestingly, we observed a slight decrease in ICP0 and gC expression in a high MOI experiment (Figure 5C) at 18 h.p.i. However, we observed, at best minor, a difference in virus yield between the individual FoxO deficient cells and the negative control at high (Figure 5D) and low (Figure 5E) MOIs. These results led us to the conclusion that individual FoxO proteins are not required for efficient HSV-1 infection. It is possible that the FoxO family protein members have, at least to some extent, redundant functions and compensate for individual depletion.

## 4. Discussion

Deregulation of the host miRNAs has been observed in a number of infections including studies involving HSV-1 [21,23,26,29,39,40,41,42,43,44,45,46,47]. However, the discrepancy between the different studies (i.e., miRNAs that were deregulated) was quite significant, indicating a strong dependence on the miRNAome of the cells used in the experiments. In this study, we aimed to investigate the deregulation of miRNAs in the primary cells, and in two completely unrelated experiments to identify a genuine miRNA response of the infected cells. We showed that miR-183, -96, -182, and -375 are the only reproducibly deregulated miRNAs in a number of primary cells, but not in all cancer-derived cells. The observed discrepancy with cancer cell lines and different studies can be explained by the fact that cancer cell lines can have dramatically altered miRNAomes [73], which probably masks the potential upregulation. Another possibility is that the pathways leading to the transcriptional activation of the cluster are lost in cancer cells (i.e., not responsive to the ICP0-mediated upregulation). A recent study by Kim et al. in which the authors analyzed the miRNA expression in tears of patients with herpes epithelial keratitis and found that among many, miR-182 and miR-183 were upregulated, supporting the idea that the activation of miR-183/96/182 represents the innate antiviral response to HSV-1 infection [47]. Similarly, the same cluster has been found upregulated in other herpesvirus infections [27] and also in completely distant RNA viruses [74,75,76]), which indicates a conserved innate antiviral response mechanism. Indeed, although the exact role of this cluster in antiviral defense is yet to be elucidated, there are multiple lines of evidence indicating an important role in the control of innate and adaptive immunity (reviewed in [77]). miRNAs of the cluster target protein phosphatase 2 catalytic subunit alpha (PPP2CA) and tripartite motif-containing 27 (TRIM27), a negative regulator of IRF3, STAT1, and TBK1 [76], promote IFN production and signaling. The enhanced IFN signaling can lead to the reduction in virus replication, as has been shown for vesicular stomatitis virus [76]. Moreover, Stittrich et al. showed that the signal transducer and activator of transcription 5 (STAT5)-mediated activation of miR-182 by interleukin 2 receptor (IL-2R) initiates a positive feedback response by targeting FoxO1 and promotes the clonal expansion of activated helper T lymphocytes [66]. Taken together, one might expect that the upregulation of the miR-183/96/182 cluster will affect HSV-1 infection; however, our results show that this is not the case, which is not surprising. In fact, HSV-1 has developed numerous mechanisms to circumvent the host antiviral defense of the cell including various functions of incoming virion/tegument proteins and proteins expressed after infection (reviewed in [78,79]). For example, only the TBK-1/IRF3/IFN axis is targeted by at least five different viral proteins, ICP0, ICP27, ICP34.5, US3, and vhs (reviewed in [78,79]). This indicates that activation of the miR-183/96/182 cluster might be part of an early cellular response to virus infection, preceding the adaptation of large DNA viruses to it. However, Oussaief et al. found that the expression of the miR-183/96/182 cluster is repressed in cells latently infected with Epstein–Barr virus (EBV mediated by EBV-encoded latent membrane protein 1 (LMP-1)) [28]. These results show that, for the virus, the roles of the miR-183/96/182 cluster in the latency program might be in contrast to productive infection and that their control might be required to enable efficient latent infection. It is difficult to predict a role for these miRNAs in the establishment and/or maintenance of latent HSV-1, particularly in the absence or limited ICP0 expression, however, this is yet to be revealed. 

On the other hand, one can argue that miRNAs cannot have a meaningful role during productive HSV-1 infection. First, productive HSV-1 infection is rather rapid, in contrast to protracted CMV replication, and inevitably leads to the destruction of the infected cell. Second, it has been shown that HSV-1 interferes with miRNA function and biogenesis at the stage of nuclear export [80,81], which limits the potential of miRNA regulation in general. Thus, it would be rather surprising that miRNAs, which accumulate late in infection, can exert a relevant control of the genes. Indeed, in our study, we show that the protein levels of members of the FoxO protein family, the confirmed targets of miRNAs of the upregulated cluster, increased early in infection followed by a decrease late in infection. The late phenomenon might be contributed to the regulation of miRNAs; however, the same observation was true in cells in which we did not detect an increase in miRNAs after infection. Thus, we can conclude that miR-183, -96, and -182, at best, have a minor contributing role in this process. Nonetheless, although we were not able to establish a hypothesized regulatory feedback loop, we were very intrigued with the expression of the FoxO protein family members. 

In this study, we showed that in cells devoid of FoxO1 or FoxO3 expression, HSV-1 replication is as efficient as it is in WT cells, indicating that individual FoxO proteins are not required for efficient replication. Potential compensatory/overlapping functions of other members might explain the lack of a phenotype, or these proteins might exert their role in latency. It is important to mention that we were not able to generate cells in which FoxO4 or all FoxO proteins were inactivated. In contrast to our observation, roles for individual FoxO proteins have been shown for EBV and KSHV. For example, it has been shown that in EBV-positive advanced nasopharyngeal carcinoma cells, which exhibit type II latency, EBV LMP1 modulates the PI3K/AKT/FoxO3 pathway, resulting in the accumulation of FoxO3 phosphorylation. This inactivation of FoxO3 led to the induction of miR-21, which in turn downregulated programmed cell death 4 (PDCD4) and Fas ligand and reduced apoptosis [82]. In addition, Liu et al. showed that in EBV-associated gastric cancer cells that showed type I/II latency, the FoxO1, FoxO3, and FoxO4 protein levels were significantly lower when compared to the control cells. Interestingly, FoxO1 mRNA was downregulated, but not FoxO3 or FoxO4 mRNAs, which might suggest diverse regulatory mechanisms by various EBV latent genes [57]. Similarly, FoxO1 has been shown to sufficiently suppress Kaposi’s sarcoma-associated herpesvirus (KSHV) lytic replication and maintain latency by controlling the levels of reactive oxygen species (ROS), since the disruption of FoxO1 could trigger KSHV reactivation and induce lytic infection by increasing the cellular ROS level [60]. On the other hand, a recent study investigating human cytomegalovirus (HCMV) has shown that FoxO1 and FoxO3 positively regulate virus lytic gene expression by activating alternative major immediate-early gene promoters and promoting IE re-expression and virus reactivation [61]. Taken together, our study shows that individual FoxO proteins are not required for efficient virus replication; however, more research, and likely, an adequate in vitro latency model, is needed to understand their roles in HSV-1 latency.

## Figures and Tables

**Figure 1 viruses-14-01661-f001:**
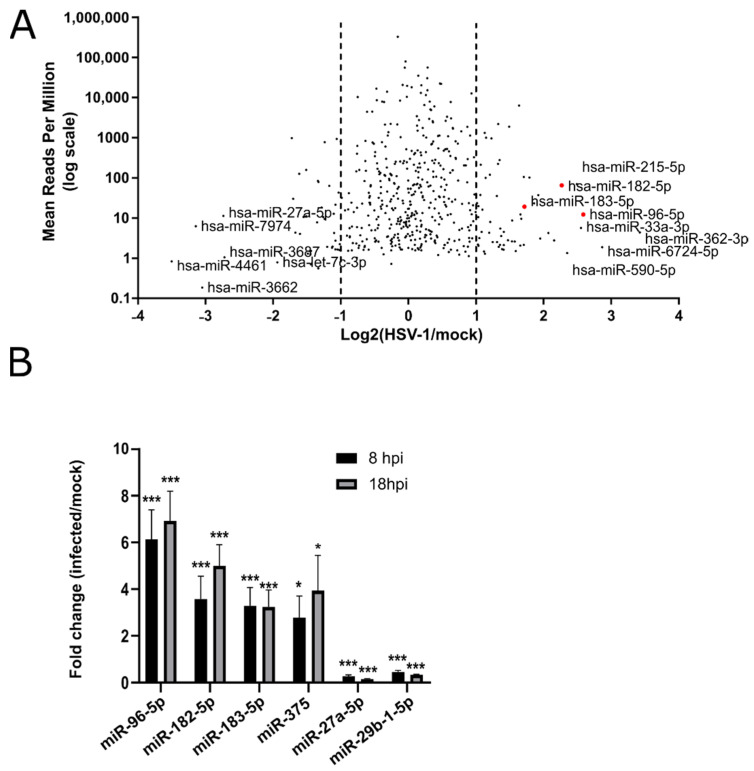
The host miRNAs are deregulated in HSV-1 infection. (**A**) Log2 value of 18 h after infection over uninfected (mock-infected) samples shown as mean of reads per million. The most deregulated miRNAs are indicated, and the position of miRNAs of the miR-183/96/182 cluster is indicated with red dots. (**B**) Fold change of miRNA expression at 8 (black bars) and 18 (gray bars) hours after infection over the mock-infected cells. Statistical significance is indicated with asterisks (* *p* < 0.05, *** *p* < 0.001).

**Figure 2 viruses-14-01661-f002:**
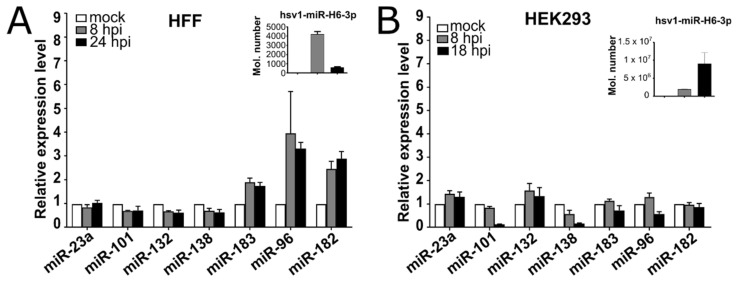
A host miRNA cluster miR-183/96/182 is upregulated in human fibroblasts. (**A**,**B**) Indicated cells (upper part of panels) were infected at an MOI of 10 and total RNA was extracted at the indicated time points. The expression levels of miRNAs were determined by RT-qPCR using stem-loop specific assays and normalized to a let-7a expression. The expression of vmiR-H6-3p is shown in the upper right corner of the panels.

**Figure 3 viruses-14-01661-f003:**
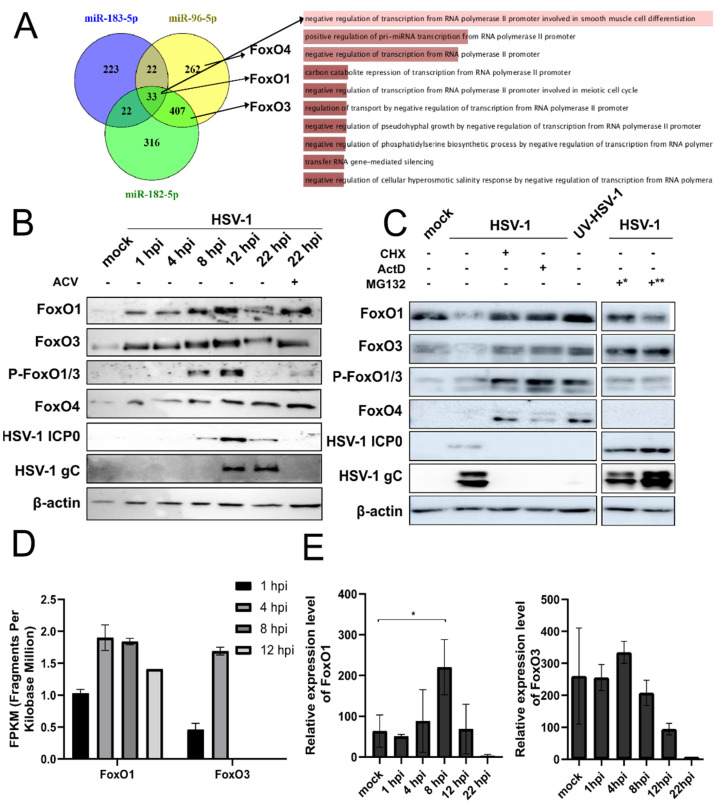
The members of the FoxO family are predicted targets of miRNAs of the miR-183/96/182 cluster and are induced during productive HSV-1 infection. (**A**) The Venn diagram shows the number of conserved targets for miR-183 (blue circle), miR-96 (yellow circle), and miR-182 (green circle). The predicted targets of the FoxO family are indicated with arrows. Gene Ontology was determined by the Enrichr Gene Ontology analysis tool (Ma’ayan Lab, Icahn School of Medicine at Mount Sinai, NY, USA) [68]. The length of the bar and the color indicate the number of genes in the different functional categories, i.e., the longer the bar and the lighter the color, the greater the number of genes. Genes with only poorly conserved sites are not shown. (**B**) The RPE cells were mock-infected (mock), infected with HSV-1 (wt), or infected and treated with acyclovir (ACV), collected at indicated time points, and analyzed by Western blot. ICP0, immediate-early viral gene, and gC, late viral gene. (**C**) HEK293 cells were mock-infected (mock), infected with HSV-1 (wt), or infected and pretreated with indicated inhibitors: cycloheximide (CHX), actinomycin D (ActD), proteasome inhibitor MG132 (*) 30 min before infection, or (**) 2 h post-infection; or infected with UV-inactivated HSV-1 (UV-HSV-1). Proteins for the Western blot were collected at 12 h.p.i. (**D**) Levels of the *FoxO1* and *FoxO3* transcripts were obtained by the total RNA sequencing and bioinformatics analysis. Results were normalized for the sequencing depth and gene length and reported in fragments per kilobase million (FPKM). (**E**) The expression levels of the *FoxO1* mRNAs was determined by RT-qPCR and normalized to 18S rRNA expression. Statistical significance is indicated with asterisks (* *p* < 0.05).

**Figure 4 viruses-14-01661-f004:**
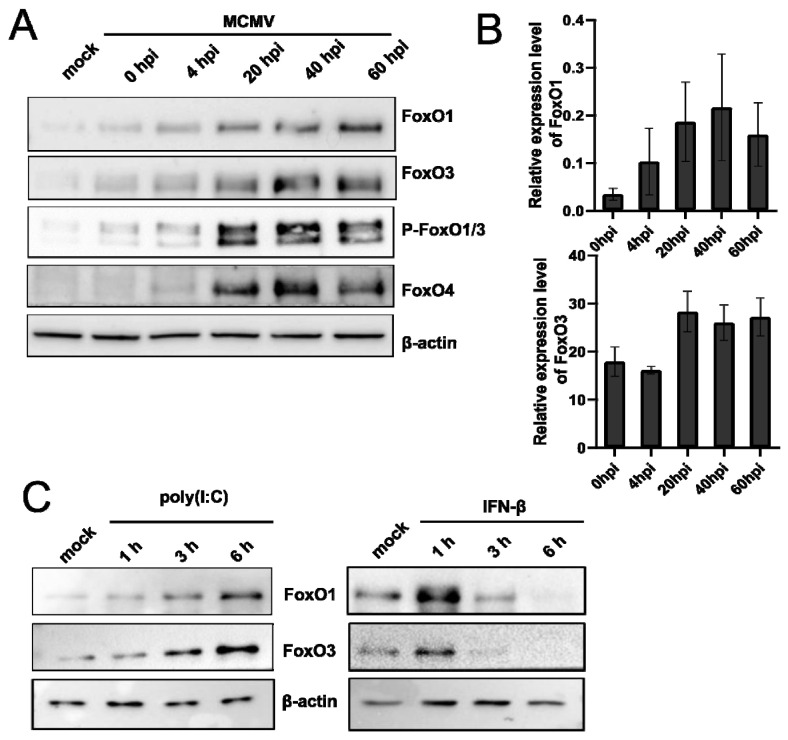
MCMV infection, IFN, and poly (I:C) induce the FoxO family of proteins. (**A**) Mouse embryonal fibroblasts (MEF) were mock-infected or infected with MCMV and samples for protein analysis were collected at the indicated time points after infection (h.p.i.). (**B**) Cells were infected as in (**A**) and RNA samples were collected. The expression levels of *FoxO1* and *FoxO3* mRNAs were determined by RT-qPCR and normalized to 18S rRNA expression. The experiment was performed in triplicate and the standard deviations are indicated. (**C**) Protein samples were collected from untreated RPE1 (mock) or cells treated with poly (I:C) (**left panel**) or IFN-β (**right panel**) at the indicated time points after treatment and analyzed by Western blot.

**Figure 5 viruses-14-01661-f005:**
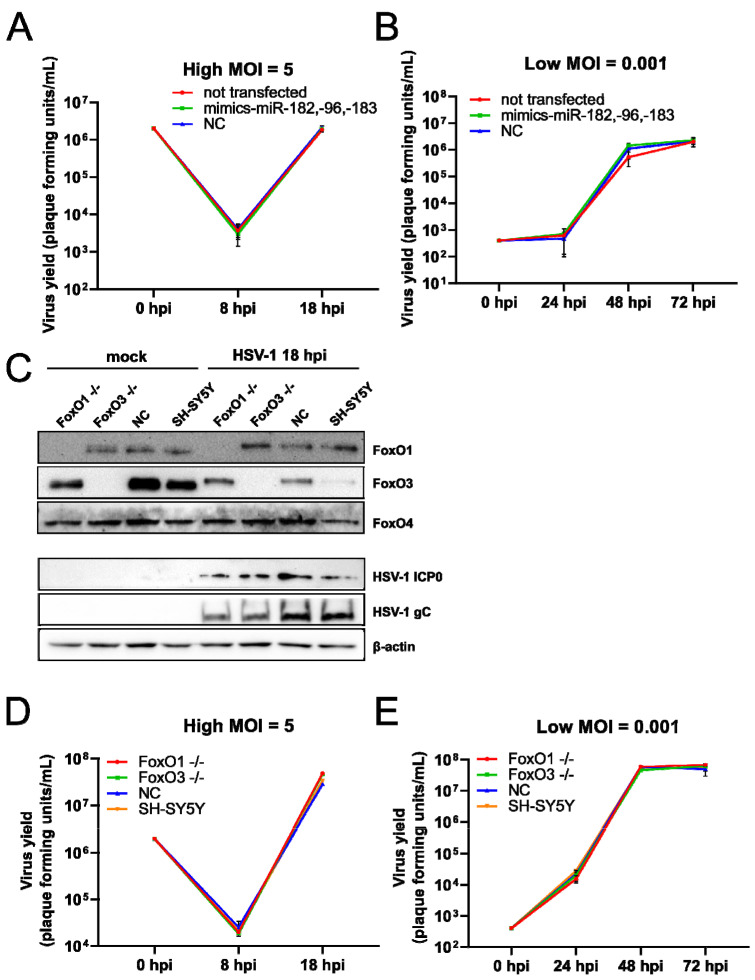
FoxO proteins are not required for HSV-1 replication. (**A**) HFFs were transfected with miR-182, -96, and -183 mimics or the negative control mimic (NC) and after 24 h infected with HSV-1 at an MOI of 5 (**A**) and 0.001 (**B**). Samples for titration were collected at the indicated time points and titrated on Vero cells. The virus yield is shown as plaque-forming units per mL (Pfu/mL). (**C**) Cells were mock-infected (mock) or infected with HSV-1 at an MOI of 5 and proteins were extracted at 18 h.p.i. for Western blot analysis. SH-SY5Y—parental cells, NC- negative control gRNA, *FoxO1−/−*, and *FoxO3−/−* cells. (**D**) Cells were infected at an MOI of 5 and 0.001 MOI (**E**) and the supernatants were collected at the indicated time points after infection (h.p.i.). The virus titer was determined by titration on Vero cells and shown as plaque-forming units per mL (Pfu/mL). 0 h.p.i. corresponds to the virus titer in the infectious medium. SH-SY5Y—parental cells, NC—negative control gRNA, *FoxO1−/−*, and *FoxO3−/−* cells.

## Data Availability

Not applicable.

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
