# Peer review of "The Virus-Induced Upregulation of the miR-183/96/182 Cluster and the FoxO Family Protein Members Are Not Required for Efficient Replication of HSV-1"

_viruses, 2022, doi:10.3390/v14081661_

Round 1

Reviewer 1 Report

The work by Zubković et al., titled 'The virus-induced upregulation of the miR-183/96/182 cluster 2 and the FoxO family protein members are not required for the 3 efficient replication of HSV-1' is overall an interesting and well executed study.

Some minor comments:

1. Have the experiments in Figure 4 repeated. If yes the graph should represent that.

2. On the same note the figure legends should mention the number of time each experiment was repeated.

Author Response

Dear Reviewer,

Thank you for reviewing our manuscript "The virus-induced upregulation of the miR-183/96/182 cluster and the FoxO family protein members are not required for efficient replication of HSV-1" by Zubković et al. We are very grateful for your kind comments and your efforts to improve our manuscript.

We have addressed all of your concerns, the description of which you will find below and which are marked "track-change" in the manuscript.
We thank you for your help

Igor Jurak

Reviewer:

The work by Zubković et al., titled 'The virus-induced upregulation of the miR-183/96/182 cluster 2 and the FoxO family protein members are not required for the 3 efficient replication of HSV-1' is overall an interesting and well executed study.

Some minor comments:

  1. Have the experiments in Figure 4 repeated. If yes the graph should represent that.

RESPONSE: Thank you very much for bringing that up. The whole experiment was repeated twice and we included a new version of figure 4. B showing the RTqPCR triplicate analysis of FoxO1 and FoxO3.

  1. On the same note the figure legends should mention the number of time each experiment was repeated.

RESPONSE: In all figures we have included, where appropriate, an explanation of a number of experiments performed.

Thank you 

Reviewer 2 Report

In this manuscript, Zubkovic and colleagues studied the expression of the miRNA cluster miR-183/96/182 following HSV-1 infection and analyzed the modulation of the FoxO family protein, which is usually regulated by the above miRNA cluster.

Despite the large amount of work, the manuscript is confusing. The strategy is not clear and/or well described. In addition, the manuscript lacks an adequate conclusion. Therefore, the manuscript should be extensively revised before publication.

Among others, these are some of the issues that need to be solved:

-It is not correct to use the expression of miR-H6 as an indicator of virus replication. Authors should perform Standard Plaque assay.

-Manuscript is confused. Authors continuously refer to unpublished data, describing results that are not included in the figures. Those parts should be removed, as they are not relevant.

-Figure references should be adjusted in the text. It is difficult to follow results’ description if it is not properly supported to fig references.

- Authors use an enormous amount of cell lines. The authors should select some of the most relevant results and focus the paper on them. It is inconsistent to have the experiments of figure 1 performed on HFF, results of figure 3B on RPE cell and results of fig 3D on HEK293.

- On the same issue, in the first paragraph the authors justify the discordant results depending on the different cells used for the experiments (line 267 page 6). I agree that there is a strong dependence on the cells used (as shown in figure 2). However, I don’t understand how the two experiments described in 3.1 paragraph are different from each other. Are they not both performed on HFF infected with KOS strain?

-The authors discussed many of the results in the result section. Those sentences should be properly included in the discussion section instead.

-In the section 3.3 it would be interesting to analyze the regulation of miR-183/96/182 following treatment with CHX, ActD and Mg132. Authors describe an upregulation of this cluster (line 346) but results are not shown.

-Panel 2B-2D could be a supplementary image. It is confounding to have so many negative results mixed in the main figures.

-Results not shown of FoxO protein analyzed for other cell lines (described at line 338) should be added as supplementary figures.

-Figure 3B: In the figure legend it is indicated that 3B is WB and rt-PCR, however panel 3B is only WB.

-Cells in 3C (RPE) are the same as fig 3B? Moreover, why in fig 3C FoxO1 and FoxO3 decrease in samples infected for 12h with HSV-1 compared to mock infected ones, while in fig 3B result is the opposite?

-In the same figure, the results of cells treated with acyclovir, despite description in the results, is missing.

-It is not clear in the 3.3 section why authors identified a specific role for IE protein ICP0, among other IE proteins.

-Since the data of transcriptome analysis from fig 3D are inconclusive, they should be eliminated from the results. It would be useful to add a simple rt-PCR for FoxO3 and FoxO4 next to the FoxO1 (fig 3E). Moreover, results described in the 3.4 paragraph (fig 3D and 3E) should be included in the paragraph 3.3.

-Fig 5: Standard Plaque Assay quantify the amount of produced virus following infection. How could the virus be produced at 0 hpi?

- Another discordant result is in fig 5C. FoxO3 decreased in infected NC compared to mock NC. This is not concordant with fig 3B.

Minor points:

-Manuscript should be profoundly revised for editing issue (an example, HFF -line 93- is the first appearance but has not the acronym specified).

-Line 245 “high MOI”…. How much? In the figure this is not specified.

-Figure 3A submitted as figure is different to the one included in the text (could be a technical problem?)

Author Response

Dear Reviewer,

Thank you for reviewing our manuscript "The virus-induced upregulation of the miR-183/96/182 cluster and the FoxO family protein members are not required for efficient replication of HSV-1" by Zubković et al.

We are very grateful for your thorough review and your suggestions for improving our manuscript.

We have addressed all your concerns, the description of which can be found below point by point and are marked "track-change" in the manuscript.

We thank you for your valuable time

Igor Jurak

  • In this manuscript, Zubkovic and colleagues studied the expression of the miRNA cluster miR-183/96/182 following HSV-1 infection and analyzed the modulation of the FoxO family protein, which is usually regulated by the above miRNA cluster.

Despite the large amount of work, the manuscript is confusing. The strategy is not clear and/or well described. In addition, the manuscript lacks an adequate conclusion. Therefore, the manuscript should be extensively revised before publication.

RESPONSE: We thank you for your criticism, and we agree that the description of the strategy of the study can be improved. We have included changes in the results section 3.1. Briefly, to identify the best miRNA candidates for downstream analysis, we compared two independent sequencing experiments. We will deposit both datasets on the SRA server as part of this study. We believe that comparing two independent experiments can help to identify reproducibly deregulated miRNAs.

2) - Among others, these are some of the issues that need to be solved:

-It is not correct to use the expression of miR-H6 as an indicator of virus replication. Authors should perform Standard Plaque assay.

RESPONSE: Thank you for this comment. We agree with the reviewer that standard plaque measures are the correct indicator of virus replication. Late genes, such as miR-H6, are indicators of late gene expression and, limited, indicator of viral replication progress. We have corrected the statement. (p7 line 300).

  • -Manuscript is confused. Authors continuously refer to unpublished data, describing results that are not included in the figures. Those parts should be removed, as they are not relevant.

RESPONSE: We thank you for this comment. We have removed 2 out of 3 “unpublished” statements and will include datasets as part of this manuscript (p6/l261 and p11/l399). The results of one of the unpublished references are from personal communication and collaboration with a coauthor D. Pan. We refer to results that are part of a larger host-wide miRNA screen that is about to be published; and therefore, cannot be presented in our manuscript. We can provide a Figure of the results for the reviewer upon request.

  • -Figure references should be adjusted in the text. It is difficult to follow results’ description if it is not properly supported to fig references.

RESPONSE: Thank you for this note, we have revised the entire manuscript and corrected.

  • - Authors use an enormous amount of cell lines. The authors should select some of the most relevant results and focus the paper on them. It is inconsistent to have the experiments of figure 1 performed on HFF, results of figure 3B on RPE cell and results of fig 3D on HEK293.

RESPONSE: Thank you for the fair review. We agree with the reviewer that the use of different cell lines in the manuscript could be confusing. Indeed, we performed protein analyses with many cell types, but the pattern of FoxO expression was consistent throughout. We have included a new supplemental figure (Supp. Fig. S5) showing the pattern of Foxo proteins in HEK, HFF, and SY5Y cells. However, we would like to keep the RPE cell in 3B because the band shift is clearly visible, which was also visible but not as obvious in the other cell lines.

  • - On the same issue, in the first paragraph the authors justify the discordant results depending on the different cells used for the experiments (line 267 page 6). I agree that there is a strong dependence on the cells used (as shown in figure 2). However, I don’t understand how the two experiments described in 3.1 paragraph are different from each other. Are they not both performed on HFF infected with KOS strain?

RESPONSE: We thank you for this comment. We have made
changes in section 3.1 that we hope will improve the explanation of our strategy and rational. As the reviewer noted, we and others have observed significant discrepancies in the host miRNAome during HSV-1 infection, which is understandable. First, we performed miRNA sequencing as part of the project, which gave us certain results about deregulated miRNAs. Then, we decided to further validate our results by comparing them with the sequencing dataset also performed with HFFs (not the same as in the first experiment; timed before the first sequencing; the results of these experiments have not been published), but with the same chemistry and platforms. Our rationale was that by comparing two independent sequencing datasets, we could identify miRNA candidates worthy of further analysis. We were not surprised that our results were consistent with the previous findings of Lutz et al. and that only a limited number of miRNAs appeared in both experiments.

  • -The authors discussed many of the results in the result section. Those sentences should be properly included in the discussion section instead.

RESPONSE: We thank you for this comment. Our intent was to describe as best we could our rationale for the experiments we performed. We have endeavored to transfer some points from the results to the discussion section.

  • -In the section 3.3 it would be interesting to analyze the regulation of miR-183/96/182 following treatment with CHX, ActD and Mg132. Authors describe an upregulation of this cluster (line 346) but results are not shown.

RESPONSE: We thank you for that comment. We did not observe upregulation of the miR-183/96/182 cluster in HEK293 cells and therefore did not analyze miRNA expression in cells treated with different inhibitors. Importantly, we show downregulation of FoxO proteins in cells in which miRNAs are not induced. The use of inhibitors suggests that viral/host gene expression is required for depletion in the late phase of infection and can be partially restored by the addition of MG132. All reagents inhibit viral replication, so not many conclusions can be drawn. However, the partial rescue by MG132 suggests proteasomal degradation, which may indicate a role for ICP0. However, we provide no evidence in this direction.

  • -Panel 2B-2D could be a supplementary image. It is confounding to have so many negative results mixed in the main figures.

RESPONSE: We agree with the reviewer that some of the results presented can be moved to the supplements. We have created a new version of Fig. 2 that includes a positive cell line (HFFs) and a negative cell line (HEK293); other results have been transfered to the Supplemntal figure S4.

  • -Results not shown of FoxO protein analyzed for other cell lines (described at line 338) should be added as supplementary figures.

RESPONSE: We have included supplementary figure S5 with the requested results.

  • -Figure 3B: In the figure legend it is indicated that 3B is WB and rt-PCR, however panel 3B is only WB.

RESPONSE: Thank you. We have corrected the error.

  • -Cells in 3C (RPE) are the same as fig 3B? Moreover, why in fig 3C FoxO1 and FoxO3 decrease in samples infected for 12h with HSV-1 compared to mock infected ones, while in fig 3B result is the opposite?

RESPONSE: Thank you for this comment. The cells in boxes 3C and 3B are HEK293 and RPE, respectively. Please see the response to comments 5) and 6). We believe that the observed differences in the kinetics of the FoxO protein patterns are due, at least in part, to the different permissiveness of the cell lines tested. However, the pattern is the same.

  • -In the same figure, the results of cells treated with acyclovir, despite description in the results, is missing.

RESPONSE: Thank you for noticing this. We have included the description of ACV the figure legend.

  • -It is not clear in the 3.3 section why authors identified a specific role for IE protein ICP0, among other IE proteins.

RESPONSE: Thank you for this comment. We agree that the statement in the 3.3. section was too strong and not justified with any evidence. We have changed the statement (line 363).

  • -Since the data of transcriptome analysis from fig 3D are inconclusive, they should be eliminated from the results. It would be useful to add a simple rt-PCR for FoxO3 and FoxO4 next to the FoxO1 (fig 3E). Moreover, results described in the 3.4 paragraph (fig 3D and 3E) should be included in the paragraph 3.3.

RESPONSE: Thank you for this comment. We agree with the Reviewer that negative results can be omitted from the figure and that RTqPCR for Foxo3 and Foxo4 would be more informative. We have Changed the Figure 3 accordingly. However, similar to the sequencing data were not able to reproducibly detect the FoxO4 transcript. We believe that merging sections 3.3. and 3.4 would make the paragraph more difficult to read.

  • -Fig 5: Standard Plaque Assay quantify the amount of produced virus following infection. How could the virus be produced at 0 hpi?

RESPONSE: Thank you for this comment. We agree that high titers at 0 h.p.i can be misleading. 0 h.p.i. correspond to virus titer in the infectious medium. We have included description in the figure legend.

  • - Another discordant result is in fig 5C. FoxO3 decreased in infected NC compared to mock NC. This is not concordant with fig 3B.

RESPONSE: We thank your comment and careful observation. The slight discrepancy between these two experiments can be attributed to slightly lower amount of proteins loaded (actin levels).

18) Minor points:

-Manuscript should be profoundly revised for editing issue (an example, HFF -line 93- is the first appearance but has not the acronym specified

-Line 245 “high MOI”…. How much? In the figure this is not specified.

-Figure 3A submitted as figure is different to the one included in the text (could be a technical problem?)

RESPONSE: Thank you for these suggestions and comments. We have made a considerable effort to revise the manuscript. High MOI has been replaced with the exact MOI. Figure 3A has been modified and resubmitted.

Round 2

Reviewer 2 Report

The manuscript was improved during the first round, the strategy is now explicated and the figures are less confusing and it is now suitable for publication.

Author Response

Dear Reviewer,
thank you very much for reviewing our manuscript and helping us to improve it.